# The Effect of the COVID-19 Pandemic on Vaccination Behaviour of Individuals over the Age of 65 Years in Turkey: Single-Centre Experience

**DOI:** 10.3390/vaccines11010034

**Published:** 2022-12-23

**Authors:** Ali Uğur Ergin, Düriye Sila Karagöz Özen, Mehmet Derya Demirağ

**Affiliations:** Samsun Research and Training Hospital, Samsun 55090, Turkey

**Keywords:** COVID-19, elderly, vaccination, vaccine acceptance

## Abstract

The aim of this study is to evaluate the awareness of individuals over 65 years of age who have had the COVID-19 vaccine at the Samsun Research and Training Hospital and to evaluate whether the COVID-19 pandemic affected the vaccination behaviour in the geriatric age group. A total of 290 people who were vaccinated against COVID-19 at the Samsun Training and Research Hospital between 16 April 2021 and 16 April 2022 and volunteered to participate in the study were included. The questionnaire form was created by the researchers. According to the national and global guidelines, the seasonal influenza vaccine, Td or Tdap vaccines (tetanus, diphtheria, and pertussis), shingles vaccine, and pneumococcal conjugate vaccine (PCV15 or PCV20), have been recommended to all adults over the age of 65. It was questioned whether the participants had the vaccines recommended for them before and after the pandemic, if they were not vaccinated, what were the reasons, and whether the COVID-19 pandemic affected the general view on vaccination in this age group. Demographic data and comorbidities were also recorded. After each response that showed that the participant was not vaccinated, reasons were investigated with new questions to find out the causes of vaccine refusal or vaccine hesitancy. Finally, all participants were asked whether they would have the relevant vaccinations when offered. It was shown that 282 (94.3%) of the 299 people who participated in the study were considering getting a regular COVID-19 vaccine from now on, while 84.3% of the participants mentioned that the COVID-19 pandemic had a positive effect on their general vaccination behaviour. While 39 (13%) people stated that their view on vaccination was not affected by the pandemic, 8 (2.7%) people stated that it was negatively affected. The most common reason about low vaccination rates before the pandemic was a lack of enough knowledge about the recommended vaccines. The pandemic increased the vaccination awareness among the adult population. We think that vaccination rates may be improved by education of the geriatric population on this subject.

## 1. Introduction

Adult vaccination targets have not been reached as they were in childhood vaccination. Hence, the Adult Vaccination Campaign in Europe ADVICE working group was established by the European Federation of Internal Medicine (EFIM) about 10 years ago to increase adult vaccination awareness [1]. ADVICE aims to improve public health by making efforts toward lifelong immunisation to prevent vaccine-preventable diseases in adults, globally [1]. As we age, our immune systems tend to weaken over time, putting us at higher risk for certain diseases. For this reason, in addition to the seasonal flu (influenza) vaccine and Td or Tdap vaccines (tetanus, diphtheria, and pertussis), the Centres for Disease Control Immunisation Practices Advisory Committee (ACIP) recommend vaccination with the shingles vaccine, which protects against shingles and the complications from the disease (recommended for healthy adults 50 years and older), and the pneumococcal conjugate vaccine (PCV15 or PCV20), to all adults over the age of 65 [2]. The new coronavirus disease (COVID-19), which was detected in the Wuhan region of China in 2019 and spread rapidly all over the world, was declared a pandemic by the WHO in 2020 [3]. Since then, despite widespread recommendations to prevent the spread of the disease and provide treatment for sick people, 6,573,968 people died all over the world, 101,203 of which were in Turkey [4]. Doubts that all deaths were ideally recorded and reported suggest that these data may be higher than stated [5].

COVID-19 vaccines had an important effect on mortality reduction due to this disease. The general population and healthcare professionals accepted that the control of the pandemic would be possible with vaccination [6]. The World Health Organisation stated that increased anti-vaccination efforts and vaccine hesitancy are among the 10 most important public health problems [7]. Vaccine hesitancy refers to the delay in the acceptance or rejection of the vaccine. Vaccine hesitancy is complex and context-specific and may vary over time, place, and vaccine [8]. In a study from our country, 8977 vaccine refusal cases were detected in 2016 (3.5%), and 14,779 cases in 2017 (5.9%; *p* < 0.001). 

It is predicted that the population aged 65 and over will gradually increase in the world. The care of this elderly population is important both economically and socially due to the accompanying chronic diseases and weakened immune systems. The role of preventive medicine in this regard draws attention. One of our duties in preventive medicine is to ensure that this age group is vaccinated. Both our clinical experience and literature studies have shown that awareness about other vaccines has increased with the COVID-19 pandemic. The aim of this study is to evaluate the awareness of individuals over 65 years of age who have had the COVID-19 vaccine and to evaluate whether the COVID-19 pandemic affected vaccination behaviour in the geriatric age group.

## 2. Materials and Methods

The study was planned as a cross-sectional descriptive study. A power calculation was performed, and the minimum number of patients recommended to be included in the study was found to be 256, with 80% power, and a 95% confidence interval, at the *p* < 0.05 significance level [9]. A total of 299 patients over the age of 65 who were vaccinated against COVID-19 at Samsun Training and Research Hospital between 16 April 2021 and 16 April 2022 and who volunteered to participate in the study were included.

The study was approved by the Health Sciences University Samsun Training and Research Hospital Non-Interventional Clinical Research Ethics Committee (file number 2021/16/4). The data were collected through a questionnaire created by the researchers. While creating the questionnaire form, previously conducted and validated questionnaire studies on the subject were examined [10,11]. No scale was used. The researchers evaluated whether the study questions were understandable by perfoming a preliminary study with 30 random participants (10% of the main group) from the study universe. Random people were selected according to the last digit of the citizen number to avoid the information bias. According to the results of this preliminary study, the wording of the questions was revised for clarity. After that, all people who were vaccinated in the study period and fit the inclusion criteria were included in the study. Inclusion criteria were as follows: being 65 years or older, having at least 1 dose of the COVID-19 vaccine at the Samsun Training and Research Hospital vaccination centre, and volunteering to participate in the study. Exclusion criteria were being under 65 years old, have been vaccinated against COVID-19 at health institutions and organisations other than the Samsun Training and Research Hospital, having received at least 1 dose of the COVID-19 vaccine at the vaccination centre of our hospital but could not be reached at the time of the survey, not volunteering to participate in the study, and patients with communication difficulties due to comorbidities such as hearing loss or neurological disease. Written informed consent was filled in and the questionnaire was completed by interviewing each participant face-to-face.

In the study, it was questioned whether the participants had the vaccines recommended for them before and after the pandemic, if they were not vaccinated, what were the reasons, and whether the COVID-19 pandemic affected the general view on vaccination in this age group. Demographic data and comorbidities were also recorded.

The questions and the summary form of the questionnaire are presented in Table 1. In this form, a 3-answer Likert scale including “yes”, “no”, and “I do not remember” options were used to determine whether the participants got the seasonal flu vaccine, tetanus vaccine, pneumococcal vaccine, and herpes zoster vaccine before and during the pandemic period. Finally, all participants were asked whether they would have the relevant vaccinations when offered. After each response that showed that the participant was not vaccinated, reasons were investigated with new questions to find out the causes of vaccine refusal or vaccine hesitancy. In this part of the questionnaire, the participant was able to choose more than one option.

### Statistical Analysis

SPSS Program version 22.0 was used for statistical analysis. Normally distributed continuous variables were expressed as mean ± standard deviation (SD), while non-normally distributed continuous variables were expressed as median (lower limit–upper limit). Categorical variables were expressed as numbers and percentages (%). Chi-square and Fisher’s exact tests were used to compare the categorical variables between groups. The McNemar test was used to compare the vaccination behaviour of the study population during the pandemic. A *p*-value below 0.05 was considered statistically significant. 

## 3. Results

A total of 299 people aged 65 and over were included in the study. The median age of the subjects included in the study was 72.7 ± 6.6 years (65–99). Demographic characteristics of the patients are presented in Table 2.

The results of the inquiry about whether the participants had been vaccinated against recommended vaccines before and after the pandemic and whether they were considering getting these vaccines after that are shared in Table 3. 

It was shown that 282 (94.3%) of the 299 people who participated in the study were considering getting a regular COVID-19 vaccine from now on, and 84.3% of the participants mentioned that the COVID-19 pandemic had a positive effect on their general vaccination behaviour. While 39 (13%) people stated that their view on vaccination was not affected by the pandemic, 8 (2.7%) people stated that their general view on vaccination after the pandemic was negatively affected. 

When the pre-pandemic flu vaccination status and the post-pandemic flu vaccination status of the people participating in the study were compared, it was found that the number of those who received influenza vaccination after the pandemic was statistically significantly lower (*p* = 0.003). The number of those who thought to get the influenza vaccine now was significantly higher than the number of those who had the influenza vaccine before the pandemic (*p* < 0.001). When the pre-pandemic and post-pandemic pneumonia vaccination status of the participants in the study was compared, no statistically significant difference was found between the two groups (*p* = 0.494). When the change in the perspective of the people participating in the study to be vaccinated against pneumonia was examined, the number of those who thought to get the pneumonia vaccine after that was statistically significantly higher than the number of those who had the pneumonia vaccine before the pandemic (*p* < 0.001). The number of people considering getting the tetanus vaccine after that was statistically significantly higher than before the pandemic (*p* < 0.001). 

The reasons for vaccine refusal are summarised in Table 4.

## 4. Discussion

In our study, the rate of those who received the influenza vaccine before the pandemic was 46.2%, while the rate of those who received the influenza vaccine after the pandemic was 38.1%. It was learned that 81.9% of the people who participated in our survey were considering getting the influenza vaccine regularly from now on. When the pre-pandemic influenza vaccination status of the participants and their post-pandemic influenza vaccination status were compared, it was found that the number of those who received influenza vaccination after the pandemic was statistically significantly lower. We think that the reason for this decrease is the insufficient influenza vaccine in our country during the pandemic period.

When the change in the perspective of the people participating in the study to be vaccinated against influenza was examined, the number of those who thought to be vaccinated against influenza afterward was found to be statistically significantly higher than the number of those who had been vaccinated against influenza before the pandemic. Influenza vaccination rates in our study were compared with other studies in Turkey and vaccination rates around the world. In a study in which 274 people aged 65 and over were included and the rates of influenza vaccination were investigated, 52 (19%) participants received regular influenza vaccination every year, and 75 (27.4%) participants in 2015–2016 or 2016–2017. It was stated that there was a flu vaccine in the season [12]. Sulis et al. conducted a study in Canada and compared influenza vaccination rates before the pandemic (between 2015 and 2018) and during the pandemic period (between April–May and September–December 2020). Accordingly, in this study, it was observed that while the rate of influenza vaccination of people aged 65–74 was 62.6% before the pandemic, the rate of vaccination increased to 70.7% during the pandemic period, and to 84.7% towards the end of 2020. The vaccination rate of people aged 75 and over was higher than the previous group, and it was found that the vaccination rate was 76.9% even before the pandemic [13].

During the pandemic period, it was observed that this rate increased to 80%, and towards the end of 2020, it was 89.1%. In this study, it was seen that the COVID-19 pandemic had a positive effect on being vaccinated against influenza [11]. Similar to the studies above, in our study, we see that the desire to be vaccinated against influenza increased after the pandemic.

When the reasons for not vaccinating before the pandemic were examined in our study, the most common reasons were: “I did not know I needed to be vaccinated” (50.9%), “My doctor did not recommend it, I would have if it had been recommended” (47.2%), and “I do not have enough information about the vaccine” (38%). When the reasons for not vaccinating after the pandemic were examined, the most common reasons were: “My doctor did not recommend it, I would have if it had been recommended” (53.5%), “I did not know that I should be vaccinated” (51.1%), and “I do not have enough information about the vaccine” (29.7%). In the above-mentioned study, which included 274 people aged 65 and over in Turkey, it was found that the most common reasons for not getting the influenza vaccine were not knowing that the vaccine was necessary (34%) and believing that the vaccine was not necessary because they were healthy (26%) [12].

In our study, similar to the studies above, reasons for not being vaccinated were found, such as a lack of sufficient knowledge about the vaccine, not believing that the vaccine could protect against the disease, and not trusting the vaccine as it may have side effects.

In our study, the rate of those who had the pneumococcal vaccine before the pandemic was 27.8%, while the rate of those who had the pneumonia vaccine after the pandemic was 30.1%. After that, the rate of those considering regular pneumococcal vaccination was 82.9%. When the pre-pandemic and post-pandemic pneumonia vaccination status of the people participating in the study was compared, no statistically significant difference was found between the two groups. In a study conducted by cancer patients in Turkey in March–November 2020, it was found that 102 (63%) of 162 cancer patients were not vaccinated against pneumococcus, although it was recommended [14]. Another recent study from our centre demonstrated that vaccination rates among end-stage renal failure patients against pneumonia and influenza were 36% and 55.8%, respectively [15]. In a study from Italy, it was shown that 15,011 participants who were 65 or older (53.4%) had received influenza vaccination during the 2019, and 3461 (12.3%) had received anti-pneumococcal vaccination [16]. These results were greater than our vaccination rates for influenza and less than those of pneumonia before the pandemic. 

When the reasons for not getting a pneumococcal vaccine before the pandemic were examined in our study, the most common reasons were: “I did not know I needed to get vaccinated” (64.2%), “My doctor did not recommend it, I would have if it had been recommended” (59.3%), and “I do not have enough information about the vaccine” (42%). When the reasons for not getting a pneumococcal vaccine after the pandemic were examined, it was seen that the most common reasons were: “My doctor did not recommend it, I would have if it had been recommended” (61.5%), “I did not know that I should be vaccinated” (56%), and “I do not have enough knowledge about the vaccine” (44%). When the reasons for not vaccinating against pneumococcus were examined, it was seen that the most common reason was “I do not have enough information about the vaccine” (45.1%). When the reasons for not vaccinating against pneumococcus were examined in the study conducted in Switzerland, the most common reason was the lack of sufficient information about the vaccine, with 64.7%. This was followed by the physician not recommending the vaccine, doubting the vaccines, refusing to be vaccinated, not having enough time, and other reasons [17]. In our study, similar to the above study, there were reasons for not having a pneumonia vaccine, such as not having enough information about the vaccine and not having been recommended by a doctor, where if it had been recommended, they would have had it done.

In our study, it was observed that the rate of shingles vaccination of the people who participated in the survey was 0.3%. In our study, the rate of participants who planned to get the shingles vaccine when it was recommended after that was determined as 70.9%. When the change in the perspective of the people participating in the study on the shingles vaccine was examined, the number of those who thought about getting the shingles vaccine after that was found to be statistically significantly higher than the number of those who had the shingles vaccine before the pandemic. In a survey conducted in our country in 2021 involving 1425 people aged 18 and over, the rate of getting the shingles vaccine was found to be 0.3% when the participants were asked about the vaccines that they had in the last 10 years. It is seen that this rate is the same as the rate of shingles vaccination in our study. Again, in the same study, it was determined that the awareness level of the shingles vaccine was 27.6%. When these people were asked whether they planned to have the shingles vaccine in the same year, it was seen that the rate of those who planned to get the shingles vaccine was 0% [18]. In the USA, it was reported that the rate of vaccination against shingles in adults aged 60 and over was 24% in 2013 [18]. In our study, the reasons for not getting the shingles vaccine of those who have never been vaccinated were: “I do not have enough information about the vaccine” (78.2%), “I did not know I should be vaccinated” (39.6%), and “My doctor did not recommend it, I would have if it had been recommended” (30%). It was observed that the most common reason for those who did not plan to be vaccinated despite the shingles vaccine being recommended was “I do not have enough information about the vaccine” (74.7%). We could not find any information about the reasons for not being vaccinated for the shingles vaccine at the age of 65 and over in the literature search. 

In our study, it was seen that the rate of tetanus vaccination in the last 10 years of the people who participated in the survey was 18.7%, and the rate of participants who planned tetanus vaccination when it was recommended from now on was 85.3%. When the change in the perspective of the people participating in the study on the tetanus vaccine was examined, the number of those who thought about getting the tetanus vaccine after that was found to be statistically significantly higher than before the pandemic. In their study, Caristia et al. asked about the vaccinations that the participants had in the last 10 years, and it was found that the rate of getting tetanus vaccination was 29.8%. When these people were asked whether they planned to have the tetanus vaccine in the same year, it was seen that the rate of those who planned to get the tetanus vaccine was 51% [19]. In the USA, it was reported that the rate of tetanus vaccination among people aged 60 and over was 56% in 2013 [20].

In our study, when the reasons for not being vaccinated against tetanus in the last 10 years were asked, these reasons were: “I did not know I should be vaccinated” (67.5%), “My doctor did not recommend it, I would if it had been recommended” (47.9%), and “I do not have enough information about the vaccine” (39.5%). When the tetanus vaccine was recommended, the most common reason for not vaccinating against tetanus was “I do not have enough information about the vaccine” (52.3%). 

We think that the most logical reason why pre-pandemic vaccination rates are lower than post-pandemic is the lack of adequate information and guidance on adult vaccination in this age group. When it was seen by health professionals and society that it was possible to prevent the effects of the pandemic with vaccines, interest in other vaccines seems to have increased. We also realised in our own clinical observations that awareness of other vaccines has increased during this period. The people older than 65 years of age were affected by the COVID-19 pandemic in various ways. Increased body weights, worsened sleep and diet, decreased physical activity, and increased smoking habits were observed [21]. Elderly people have had high COVID-19 mortality since the beginning of the pandemic, which is why lockdown rules were applied more strictly to elderly individuals. Elderly individuals, whose lives were limited during the lockdown period, started to take part in society again when they were vaccinated. Therefore, we think that the perspective on general vaccines in this age group has also been positively affected.

The Coronavirus Disease 2019 pandemic has led to significant disruption of routine medical services. Even when medical services were available, people were unable to access them due to transportation disruptions, economic difficulties, and fear of exposure to SARS-CoV-2, and a reluctance to leave home or go to medical facilities due to precautions [20]. According to the WHO data, a 70% reduction in routine immunisation services has been reported [22]. An early indication that immunisation services have been affected has been changed to routine vaccine orders by national or regional authorities. Compared to the 2019 models, vaccine orders fell early in the pandemic, with significantly lower orders in the USA and Europe from mid-March 2020 to mid-April 2020 [23,24]. Özdemir et al. investigated the reasons why cancer patients did not receive the pneumococcal vaccine, and they showed that the most common reason why cancer patients were not vaccinated during the study was not being able to go to a health institution due to COVID-19 (*n* = 35, 34.3%) and not being able to reach the vaccine (*n* = 21, 20.6%) [14]. In our study, when the rates of influenza vaccination before and after the pandemic in our country were examined, it was observed that the rate of vaccination was higher in the pre-pandemic period and lower in the post-pandemic period. Although people wanted to be vaccinated against influenza, a sufficient number of vaccines could not be obtained to meet this demand during the study period in our country. During the study period, there was an influenza vaccine supply problem that did not exist in previous years in Turkey. Therefore, we think that post-pandemic vaccination rates are lower than in previous years.

It was learned that 94.3% of the people who participated in the study would like to be regularly vaccinated against COVID-19 when recommended from now on. It was observed that the most common reason chosen by the participants who did not want to be vaccinated against COVID-19 was “I do not trust the vaccine as it may have side effects” (58.8%). In a recent study, which included 467 people, examining the psychosocial factors affecting the COVID-19 vaccine acceptance in Turkey, it was seen that majority of the people (44.1%) who participated in the survey trusted the positive effects of the vaccine and wanted to be vaccinated. It was observed that 40.9% of them did not trust the positive effects of the vaccine but still wanted to be vaccinated [23]. In our study, similar to the studies mentioned above, when it was recommended to be regularly vaccinated for COVID-19, it was seen that the most common reason for not being vaccinated was distrust of the vaccine due to side effects. Another recent study from our clinic investigated the rates of COVID-19 vaccine acceptance among pregnant women during the pandemic, and the most common reason for vaccine refusal was the fear of side effects, the same as in this study [25].

As a result, in our study, the number of those who plan to have these vaccines for influenza, pneumococcal, tetanus, and shingles was found to be statistically significantly higher than the number of those who had these vaccines before the pandemic. It was seen that 84.3% of the people who participated in our study had a positive outlook on the general view of vaccination of the COVID-19 pandemic. Supporting our study, in a cross-sectional survey study conducted by the Hacettepe University vaccination centre, including 1009 patients between 13 and 20 April 2021, 44.5% of the people participating in the study said that the COVID-19 pandemic positively affected their view of vaccination, 27.5% of them did not want to be vaccinated, and 28% of them stated that they were undecided about the subject [24].

The strength of our study is that it contributes to the literature by revealing the vaccination rates of the COVID-19 pandemic in the geriatric group. Again, in our country, there are not enough studies showing the vaccination behaviour of individuals over the age of 65 regarding all vaccines, and our study also contributes to the literature. The data we obtained reveal the reasons for vaccine hesitancy in individuals over 65 years of age, for each vaccine. However, the study was conducted in people who had the COVID-19 vaccine. The lack of a control group that refused to be vaccinated against COVID-19 is among the limitations of our study. Another limitation of our study is that it was conducted with a group of patients who applied to a single centre. We think that multicentre studies are needed to generalise the findings to the whole population.

## 5. Conclusions

We think that the rate of influenza and pneumococcal vaccination of people after the pandemic is lower than expected due to the general lockdown applied during the period and the decrease in people’s access to health institutions due to the fear of contamination. It was observed that the people participating in our study have a positive outlook on the general view of vaccination of the COVID-19 pandemic. The most common reason for low vaccination rates before the pandemic was a lack of enough knowledge about the recommended vaccines. The pandemic increased the vaccination awareness among the adult population. We think that vaccination rates may be improved by education of the geriatric population on this subject. Furthermore, we believe that the data we obtained in this study will be a guide in taking precautions against vaccine hesitancy.

## Figures and Tables

**Table 1 vaccines-11-00034-t001:** The questionnaire form.

According to Turkish Infectious Diseases and Clinical Microbiology Specialization Association (EKMUD); Recommended Vaccines over the Age of 65: Influenza (Inactivated) Vaccine is Recommended Once a Year. Tetanus, Diphtheria, Pertussis; It Is Recommended to Have it Done in Td form Every 10 years. VaricellaZoster (Recombinant) Vaccine Is Recommended to Be AdministeredAs 1 Dose over 60 Years of Age. Pneumococcal Vaccine, Polysaccharide Vaccine (PPSV23) and Conjugate Vaccine (PCV13) is Recommended to be Administered Once.	Name and Surname: Citizen Number: Date of Birth: Gender: Education Status: Comorbid Disease:
Below are 11 questions. This survey measures the impact of individuals over the age of 65 who have been vaccinated against COVID 19 on their thought on vaccination. Please tick one of the following answers from “YES”, “NO” or “I DON’T KNOW” to the most appropriate one for you.	
QUESTION 1: Did you have an influenza (flu) vaccine before the pandemic?	Yes No I don’t know
If your answer is NO or I DON’T KNOW, please tick the appropriate reasons. You can tick more than one option.	1. I do not trust the vaccine as it may have side effects. 2. I do not believe that the vaccine can protect me from the disease. 3. I do not have enough information about the vaccine. 4. I did not know that I should be vaccinated. 5. I encountered side effects when I was vaccinated before. 6. I do not want to have it done because of the news in the media. 7. I don’t want to have it done for religious reasons. 8. My doctor did not recommend it, if he did, I would have it done. 9. I think that alternative and complementary medicine methods are more effective than vaccines and have fewer side effects.
QUESTION 2: Have you been vaccinated against influenza (flu) after the pandemic?	Yes No I don’t know
If your answer is NO or I DON’T KNOW, please tick the appropriate reasons. You can tick more than one option.	Same as above
QUESTION 3: Do you plan to get an influenza (flu) vaccine from now on?	Yes No I don’t know
If your answer is NO or I DON’T KNOW, please tick the appropriate reasons. You can tick more than one option.	Same as above
QUESTION 4: Have you been vaccinated against tetanus in the last 10 years?	Yes No I don’t know
If your answer is NO or I DON’T KNOW, please tick the appropriate reasons. You can tick more than one option.	Same as above
QUESTION 5: If it is recommended to you from now on, would you get the tetanus vaccine, which should be given every 10 years?	Yes No I don’t know
If your answer is NO or I DON’T KNOW, please tick the appropriate reasons. You can tick more than one option.	Same as above
QUESTION 6: Have you ever been vaccinated against shingles?	Yes No I don’t know
If your answer is NO or I DON’T KNOW, please tick the appropriate reasons. You can tick more than one option.	Same as above
QUESTION 7: If it is recommended to you from now on, would you be vaccinated against shingles?	Yes No I don’t know
If your answer is NO or I DON’T KNOW, please tick the appropriate reasons. You can tick more than one option.	Same as above
QUESTION 8: Have you been vaccinated against pneumonia (pneumococcal 13 or 23) before the pandemic?	Yes No I don’t know
If your answer is NO or I DON’T KNOW, please tick the appropriate reasons. You can tick more than one option.	Same as above
QUESTION 9: Have you been vaccinated against pneumonia (pneumococcal 13 or 23) after the pandemic?	Yes No I don’t know
If your answer is NO or I DON’T KNOW, please tick the appropriate reasons. You can tick more than one option.	Same as above
QUESTION 10: Do you get the pneumonia vaccine (pneumococcal 13 or 23) regularly from now on?	Yes No I don’t know
If your answer is NO or I DON’T KNOW, please tick the appropriate reasons. You can tick more than one option.	Same as above
QUESTION 11: You have been vaccinated against COVID 19 in the pandemic. After that, if you are offered regular vaccination for COVID 19, would you do it?	Yes No I don’t know
If your answer is NO or I DON’T KNOW, please tick the appropriate reasons. You can tick more than one option.	Same as above
QUESTION 12: How did the COVID 19 pandemic affect your opinion on vaccination?	1. It didn’t affect my opinion on getting vaccinated. 2. From now on, it had a positive effect, causing me to have more regular vaccinations. 3. After that, it had a negative impact on not being vaccinated.

**Table 2 vaccines-11-00034-t002:** Demographical properties of the participants.

	N	(%)
**Gender**		
Female	141	47.2
Male	158	52.8
**Educational status**		
Literate	18	6.0
Primary school graduate	160	53.5
Secondary school graduate	18	6.0
High school graduate	26	8.7
Graduated from a university	21	7.0
**Comorbidities**		
Diabetes mellitus	94	31.4
Hypertension	187	62.5
Malignancy	15	5.0
Serebrovascular disease	13	4.3
Cardiac disease	107	35.8
Chronic kidney disease	48	16.1
Other	94	31.4

**Table 3 vaccines-11-00034-t003:** Vaccination attitudes of the participants before and after the pandemic.

	Yes	No	I Do Not Remember
	*n* (%)	*n* (%)	*n* (%)
Did you get vaccinated against influenza before the pandemic?	138 (46.2)	155 (51.8)	6 (2.0)
Did you get vaccinated against influenza after the pandemic?	114 (38.1)	181 (60.5)	4 (1.3)
Are you considering getting the flu vaccine from now on?	245 (81.9)	33 (11.0)	21 (7.0)
Did you get the tetanus vaccine in the last 10 years?	56 (18.7)	227 (75.9)	16 (5.4)
If the tetanus vaccine is recommended after this, would you get it?	255 (85.3)	26 (8.7)	18 (6.0)
Have you had the shingles vaccine before?	1 (0.3)	287 (96)	11 (3.7)
If the shingles vaccine is recommended after this, would you get it?	212 (70.9)	47 (15.7)	40 (13.4)
Did you get the pneumonia vaccine before the pandemic?	83 (27.8)	209 (69.9)	7 (2.3)
Have you been vaccinated for pneumonia after the pandemic?	90 (30.1)	206 (68.9)	3 (1.0)
Would you consider getting the pneumonia vaccine from now on?	248 (82.9)	26 (8.7)	25 (8.4)

**Table 4 vaccines-11-00034-t004:** The reasons for not getting vaccinated.

I Do Not Trust the Vaccine as it May Have Side Effects.	I Do Not Believe That the Vaccine Can Protect Me from the Disease.	I Do Not Have Enough Information About the Vaccine.	I Did Not Know That I Should Be Vaccinated.	I Encountered Side Effects When I Was Vaccinated Before.	I Do Not Want to Have It Done Because of the News in the Media	I Do Not Want to Have it Done for Religious Reasons.	My Doctor Did Not Recommend it, If He Did, I Would Have It Done.	I Think That Alternative and Complementary Medicine Methods Are More Effective Than Vaccines and Have Fewer Side Effects.		
17 (10,6)	17 (10,6)	62 (38,5)	82 (50,9)	1 (0,6)	2 (1,2)	0 (0)	76 (47,2)	9 (5,6)	*n* (%)	Reasons for not getting an influenza (flu) vaccine before the pandemic
19 (10,3)	20 (10,8)	55 (29,7)	94 (51,1)	2 (1,1)	3 (1,6)	0 (0)	99 (53,5)	8 (4,3)	*n* (%)	Reasons for not getting an influenza (flu) vaccine after the pandemic
18 (33,3)	19 (35,2)	20 (37)	6 (11,1)	3 (5,6)	1 (1,9)	0 (0)	10 (18,5)	3 (5,6)	*n* (%)	Reasons for not getting an influenza (flu) vaccine after that
15 (6,2)	7 (2,9)	96 (39,5)	164 (67,5)	5 (2,1)	3 (1,2)	2 (0,8)	116 (47,9)	2 (0,8)	*n* (%)	Reasons for not getting a tetanus vaccine in the last 10 years
10 (22,7)	8 (18,2)	23 (52,3)	6 (13,6)	0 (0)	0 (0)	0 (0)	8 (18,6)	0 (0)	*n* (%)	Reasons for not getting the tetanus vaccine, which should be given every 10 years if it is recommended to you from now on
17 (5,7)	8 (2,7)	233 (78,2)	118 (39,6)	1 (0,3)	2 (0,7)	1 (0,3)	89 (30)	1 (0,3)	*n* (%)	Reasons for not getting the shingles vaccine until now
17 (19,5)	8 (9,2)	65 (74,7)	18 (20,7)	0 (0)	0 (0)	0 (0)	17 (19,5)	0 (0)	*n* (%)	Reasons for not getting the shingles vaccine if recommended to you from now on
10 (4,6)	5 (2,3)	92 (42,6)	138 (64,2)	2 (0,9)	2 (0,9)	0 (0)	128 (59,3)	0 (0)	*n* (%)	Reasons for not getting a pneumonia vaccine (pneumococcal 13 or 23) before the pandemic
15 (7,2)	8 (3,8)	92 (44)	117 (56)	0 (0)	2 (1)	1 (0,5)	128 (61,5)	1 (0,5)	*n* (%)	Reasons for not getting a pneumonia vaccine (pneumococcal 13 or 23) after the pandemic
15 (29,4)	10 (19,6)	23 (45,1)	11 (21,6)	0 (0)	2 (3,9)	0 (0)	9 (17,6)	1 (2)	*n* (%)	Reasons for not regularly receiving the pneumonia vaccine (pneumococcal 13 or 23) after that
**10 (58,8)**	5 (29,4)	6 (35,3)	1 (5,9)	3 (17,6)	2 (11,8)	0 (0)	1 (5,9)	0 (0)	*n* (%)	You got the COVID-19 vaccine in the pandemic. If you are subsequently offered regular vaccination for COVID-19, reasons not to

## Data Availability

The data presented in this study are available upon request from the corresponding author.

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
