# Peer review of "The Effect of the COVID-19 Pandemic on Vaccination Behaviour of Individuals over the Age of 65 Years in Turkey: Single-Centre Experience"

_vaccines, 2022, doi:10.3390/vaccines11010034_

Round 1

Reviewer 1 Report

The submitted paper requires some modifications. In the title it has to be included what population it refers. Do your results show the opinion of the European nation or maybe one country?

I suggest to add more papers refering to COVID published in this journal and the other ones of MDPI to show that it matches to the scope:

Gallè, F., Sabella, E. A., Roma, P., Ferracuti, S., Da Molin, G., Diella, G., ... & Napoli, C. (2021). Knowledge and lifestyle behaviors related to COVID-19 pandemic in people over 65 years old from southern Italy. International Journal of Environmental Research and Public Health, 18(20), 10872.

Noale, M., Trevisan, C., Maggi, S., Antonelli Incalzi, R., Pedone, C., Di Bari, M., ... & Group, O. B. O. T. E. W. (2020). The association between influenza and pneumococcal vaccinations and SARS-Cov-2 infection: Data from the EPICOVID19 web-based survey. Vaccines, 8(3), 471.

Filho, S. Á., Ávila, J. S., Mrugalska, B., de Souza, N. F., Gomes de Carvalho, A. P. M., & Gonçalves, L. R. (2021). Decision Making in Health Management during Crisis: A Case Study Based on Epidemiological Curves of China and Italy against COVID-19. International Journal of Environmental Research and Public Health, 18(15), 8078.

The begining of the abstact and conclusions are the same. Please revise.

Author Response

Dear reviewer,

Below are our answers to your suggestions. The revisions are highlighted in green in the text.

The submitted paper requires some modifications. In the title, it has to be included what population it refers to. Do your results show the opinion of the European nation or maybe one country?

The title has been changed to “The effect of COVID-19 pandemic on vaccination behavior of individuals over the age of 65 years in Turkey: single center experience

I suggest adding more papers referring to COVID published in this journal and the other ones of MDPI to show that it matches the scope:

Answer: The following references are added.

Gallè, F., Sabella, E. A., Roma, P., Ferracuti, S., Da Molin, G., Diella, G., ... & Napoli, C. (2021). Knowledge and lifestyle behaviors related to the COVID-19 pandemic in people over 65 from southern Italy. International Journal of Environmental Research and Public Health18(20), 10872.

Noale, M., Trevisan, C., Maggi, S., Antonelli Incalzi, R., Pedone, C., Di Bari, M., ... & Group, O. B. O. T. E. W. (2020). The association between influenza and pneumococcal vaccinations and SARS-Cov-2 infection: Data from the EPICOVID19 web-based survey. Vaccines8(3), 471.

Filho, S. Á., Ávila, J. S., Mrugalska, B., de Souza, N. F., Gomes de Carvalho, A. P. M., & Gonçalves, L. R. (2021). Decision Making in Health Management during Crisis: A Case Study Based on Epidemiological Curves of China and Italy against COVID-19. International Journal of Environmental Research and Public Health18(15), 8078.

The beginning of the abstract and conclusions are the same. Please revise.

Answer: It has been revised in line with your suggestions

Reviewer 2 Report

My comment

Abstract

Abstract needs to be improved.

R1. Author don’t need to add the sentences to explain the aim of the study. Please remove the explanation related to the aim of the study, keep only the aim of the study.

R2. In the abstract the author did not show the study design, and the setting of your study. Please add in your manuscript the study design and where the study was carried out.

R3. Author needs to add the conclusion and recommendation in the abstract.

Main text

Methods section

Methods need to be improved

R4. Author needs to show the sample size calculation. How did you find 256 participants?

R5. Author needs to add the inclusion and exclusion criteria of the study.

R6. Author needs to add more explanations related to how and who the questionnaires were administrated for avoiding the information bias.

Discussion

Discussion section need to be improved

R7. Author needs to add the plausible explanations, why for almost the vaccines the rate before pandemic was less than during and after pandemic?.

R8. When the reasons for not vaccinating before the pandemic were examined in this study, the most common reasons were 'I didn't know I needed to be vaccinated' (50.9%), 'My doctor didn't recommend it, I would have if it had been recommended (47.2%) and 'I don't have enough information about the vaccine' (38%, 5) has been found. Author needs to add the recommendation in the conclusion section to improve this lack of information.

R9.  Author needs to add the strength and limitations of the study and propose the way to improve or to resolve the limitations.

R10. Regarding the reason of the weak rate of influenza vaccine after pandemic, I think there is a confusion between the discussion section and the conclusion (page 7:143-147, page 10:305-310). Please light this situation.

Conclusion

Author needs to improve the conclusion

R11. Author needs to show only the main findings and recommendation in the conclusion

Author Response

Dear reviewer,

We have made revision in line with your suggestions. Below are the answers to your comments. The revisions are highlighted in yellow in the text.

Abstract needs to be improved.

R1. Author don’t need to add the sentences to explain the aim of the study. Please remove the explanation related to the aim of the study, keep only the aim of the study.

Answer: The explanatory sentences about the aim of the study were removed from the abstract section.

R2. In the abstract the author did not show the study design, and the setting of your study. Please add in your manuscript the study design and where the study was carried out.

Answer: the study design and setting were added.

R3. Author needs to add the conclusion and recommendation in the abstract.

Answer: Conclusion and recommendation sentences were added in line with your suggestions.

Main text

Methods section

Methods need to be improved

R4. Author needs to show the sample size calculation. How did you find 256 participants?

Answer: We use two distinct programme to calculate sample size. One of them is G power and the other is https://clincalc.com/stats/samplesize.aspx. The studies investigating the influenza vaccination rates were accepted as “known population percent”, while expected rate of vaccination in our group is accepted based on similar studies in our country.

R5. Author needs to add the inclusion and exclusion criteria of the study.

Answer: The inclusion and exclusion criteria were added.

R6. Author needs to add more explanations related to how and who the questionnaires were administrated for avoiding the information bias.

Answer: All people who were vaccinated in the study period and fit the inclusion criteria were included in the study. Randomization was done for the preliminary test only.  Random people were selected according to the last digit of the citizen number to avoid the information bias. This sentence was added.

Discussion

Discussion section need to be improved

R7. Author needs to add the plausible explanations, why for almost the vaccines the rate before pandemic was less than during and after pandemic?.

Answer: “We think that the most logical reason why pre-pandemic vaccination rates are lower than post-pandemic is the lack of adequate information and guidance on adult vaccination in this age group. When it is seen by health professionals and the society that it is possible to prevent the pandemic with vaccines, interest in other vaccines seems to have increased. We also realize in our own clinical observations that awareness of other vaccines has increased during this period. Elderly people have high COVID-19 mortality since the beginning of the pandemic and that’s why lockdown rules were applied more strictly to elderly individuals.” These sentences were added to the discussion section.

R8. When the reasons for not vaccinating before the pandemic were examined in this study, the most common reasons were 'I didn't know I needed to be vaccinated' (50.9%), 'My doctor didn't recommend it, I would have if it had been recommended (47.2%) and 'I don't have enough information about the vaccine' (38%, 5) has been found. Author needs to add the recommendation in the conclusion section to improve this lack of information.

Answer: Recommendation added to the conclusion section in line with your suggestions.

R9.  Author needs to add the strength and limitations of the study and propose the way to improve or to resolve the limitations.

Answer: The strength of our study is that it contributes to the literature by revealing the vaccination rates of the COVID-19 pandemic in the geriatric group. Again, in our country, there are not enough studies showing the vaccination behavior of individuals over the age of 65 with regard to all vaccines, and our study also contributes to the literature. The data we obtained from here reveal the reasons for vaccine hesitancy in individuals over 65 years of age, for each vaccine.We believe that the data we obtained from here will be a guide in taking precautions against vaccine instability. However, the study was conducted in people who had the COVID-19 vaccine. The lack of a control group that refused to be vaccinated against COVID-19 is among the limitations of our study. Another limitation of our study is that it was conducted with a group of patients who applied to a single center. We think that multicenter studies are needed in order to generalize the findings to the whole population.

R10. Regarding the reason of the weak rate of influenza vaccine after pandemic, I think there is a confusion between the discussion section and the conclusion (page 7:143-147, page 10:305-310). Please light this situation.

Answer: Although people wanted to be vaccinated against influenza, sufficient number of vaccines could not be obtained to meet this demand during the study period in our country. During the study period, there was an influenza vaccine supply problem that did not exist in previous years in Turkey. Therefore, we think that post-pandemic vaccination rates are lower than in previous years.

Conclusion

Author needs to improve the conclusion

R11. Author needs to show only the main findings and recommendation in the conclusion

Answer: the conclusion section has been improved.

Round 2

Reviewer 2 Report

Thank so much for answering to my comment and to improve the manuscript.  Good luck